

# Recent rift formation and impact on the structural integrity of the Brunt Ice Shelf, East Antarctica

Jan De Rydt[1], G. Hilmar Gudmundsson[1], Thomas Nagler[2], Jan Wuite[2], and Edward C. King[1]

[1]British Antarctic Survey, High Cross, Madingley Road, CB3 0ET, Cambridge, UK
[2]ENVEO, ICT-Technologiepark, Technikerstr. 21a, 6020 Innsbruck, Austria

*Correspondence to:* Jan De Rydt (janryd69@bas.ac.uk)

**Abstract.** We report on the recent reactivation of a large chasm in the Brunt Ice Shelf, East Antarctica, in December 2012, and the formation of a 50-km long new rift in October 2016. Observations from a suite of ground based and remote sensing instruments between January 2000 and July 2017 were used to track progress of both cracks in unprecedented detail. Results reveal a steady accelerating trend in the widening of the rifts, in combination with alternating episodes of fast (> 600m/day) and slow propagation of the crack tip, controlled by the heterogeneous structure of the ice shelf. A numerical ice-flow model and a simple fracture propagation criterion were successfully used to hindcast the observed trajectories, and to simulate future rift progression under different assumptions, showing a high likelihood of ice loss at the McDonald Ice Rumples, the only pinning point of the ice shelf. The nascent iceberg calving and associated reduction in pinning of the Brunt Ice Shelf may provide a uniquely monitored natural experiment of ice shelf variability, and provoke a deeper understanding of similar processes elsewhere in Antarctica.

## 1 Introduction

The Brunt Ice Shelf (BIS) is located along the Caird Coast at the eastern edge of Coats Land in East Antarctica, shown in Figure 1. It is a dynamic environment, characterised by alternating decades of fast and slow flow (Thomas, 1973; Simmons and Rouse, 1984a, b; Simmons, 1986; Gudmundsson et al., 2017) and a number of large rifts that penetrate the full thickness of the ice shelf, and stretch deep into its interior (Anderson et al., 2014). These rifts are indicators of historical and recent glaciological activity, and are long-lived features that form as the ice shelf goes through phases of steady growth followed by rapid ice loss through calving. Observational evidence for the episodic changes in ice shelf configuration has been obtained from a unique 57-year long velocity record measured at several Halley Research Stations that have occupied locations on the BIS since 1956 (Gudmundsson et al., 2017), and from regular shipborne and satellite-derived outlines of the icefront dating back to 1915 (Anderson et al., 2014).

Using an ice dynamics model to investigate the relationship between flow velocities and ice shelf geometry, Gudmundsson et al. (2017) concluded that the observed changes in velocities can be explained by a calving-induced (un)pinning of the ice shelf at the McDonald Ice Rumples (MIR, Figure 1). The MIR are a small grounded area in the north-eastern corner of the BIS where the bottom of the ice shelf makes contact with a local rise in the seafloor bathymetry, and the resulting friction exerts



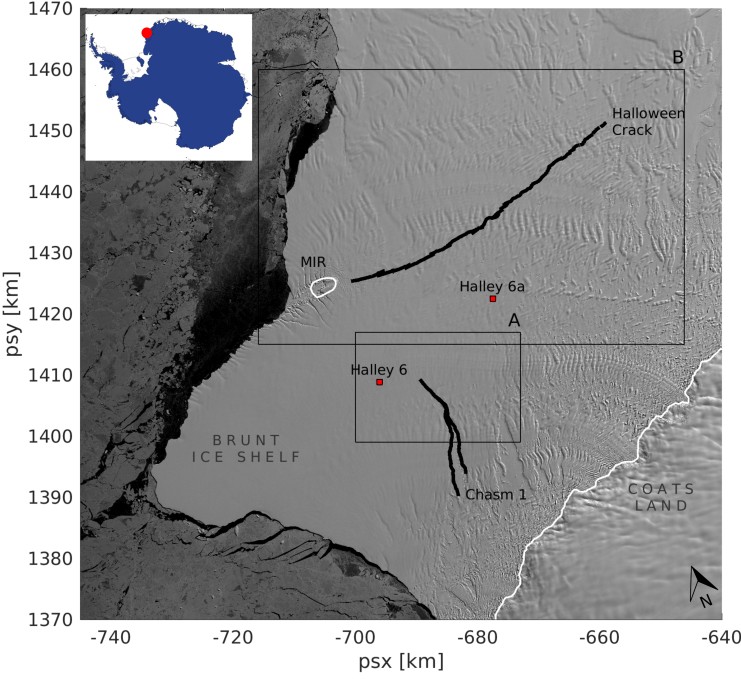

**Figure 1.** The Brunt Ice Shelf covers a 6500 km$^2$ area along the Coats Land coastline in East Antarctica (see inset). The grounding line from the SCAR Antarctic Digital Database 6.0 (www.add.scar.org) is delineated in white, and marks the boundary between the ice shelf and the adjacent continent, as well as a small pinning point at the McDonald Ice Rumples (MIR), which is absent in the ADD. The background image is a subset of a Landsat-8 scene (panchromatic band) from 15 March 2017, which was used to outline the present-day active rifts, Chasm 1 and Halloween Crack, which are highlighted in black. Red dots indicate the location of the Halley VI Research Station (operational since 2012) and its new location at Halley VIa (since February 2017). The black boxes indicate the geographical extent of Figures 2a (A) and 3 (B).

a buttressing force on the remainder of the ice shelf. This is the only known pinning point on the BIS and the much larger neighbouring Stancomb Wills Glacier Tongue, which at present cover a combined area of about 33000 km$^2$ .

Following the most recent calving event of the BIS in the early 1970s (Thomas, 1973), satellite images and aerial photography suggested that a significant part of the ice shelf in contact with the bedrock at the MIR was lost, which resulted in
5  a loss of buttressing and a two-fold increase in flow velocities between the late 1960s and the 1980s (Simmons and Rouse, 1984a). In subsequent decades, the BIS gradually readvanced and reestablished contact with the bedrock at the MIR, whilst velocities decreased to pre-1970 levels (Gudmundsson et al., 2017). This cycle of internal dynamical change controlled by calving induced unpinning and subsequent regrounding at the MIR is arguably not a unique phenomenon, as many ice shelves around Antarctica, and in particular in Dronning Maud Land, are controlled by the presence of one or several pinning points
10  (Matsuoka et al., 2015; Fürst et al., 2015; Favier et al., 2016; Berger et al., 2016). However, the long observational record and the relatively short calving frequency on the order of decades, makes the BIS an ideal location to study this cycle.



Episodic observations of the configuration of the BIS since 1915, and effectively continuous observations since the 1990s suggest that at present, the ice front is in its most advanced position since the start of measurements, and Anderson et al. (2014) hypothesized that a new calving cycle is likely to happen before 2020. In agreement with this prediction, a number of recent events have provided strong evidence for an imminent calving, in particular the growth of an old chasm that has

remained unchanged since the 1970s (Chasm 1 in Figure 1), and the rapid formation and propagation of a new rift close to the MIR in austral summer 2016 (Halloween Crack in Figure 1). In concurrence with these changes, Gudmundsson et al. (2017) reported on a new phase of ice-shelf wide acceleration as surface velocities have increased by up to 10% per year since 2012. Detailed measurements of these changes have been facilitated by the presence of the Halley VI Research Station, now located at Halley VIa (see Figure 1), and an intensive monitoring programme that has been put in place to collect important

glaciological data to help understand the dynamics of the BIS as it changes. In this study, we present the most recent findings of this programme, based on a suite of ground-based and remote sensing observations between 2000 and 2017. In particular, we report on the recent lengthening and widening of Chasm 1 since 2012, and the formation of Halloween Crack in October 2016, and provide some of the most detailed and complete observations of Antarctic rifts to date.

As both rifts are a potential precursor to iceberg calving and future widespread changes to the dynamics of the BIS, sustained

monitoring and a deeper understanding of these changes is of great relevance for many other ice shelves in Antarctica. Since iceberg formation is an important and often dominant component of their mass balance (Depoorter et al., 2013), and the weakening and total loss of ice shelves around the margins of the Antarctic Peninsula and the West Antarctic Ice Sheet have led to widespread dynamic changes to the mass balance of the ice sheet (Pritchard et al., 2012), lessons learned about the physical processes that control the dynamics of the BIS are directly transferrable to these ice shelves. Moreover, the wealth

of observational data gathered on Chasm 1 and the Halloween Crack provides a unique opportunity to advance and validate numerical models of fracture propagation, and improve the treatment of iceberg calving laws in such models.

As a first step towards this goal, a simple but successful algorithm to simulate the propagation direction of fractures was used in conjunction with a finite-element ice flow model of the BIS. Model predictions were successfully tested against observations, and used to estimate the future trajectory of both Chasm 1 and Halloween Crack, assuming continued propagation. This will

ultimately allow us to predict future changes in ice shelf extent and investigate resulting dynamic changes to the flow, along lines similar to previous studies (Khazendar et al., 2009; Larour et al., 2014; Gudmundsson et al., 2017).

In Sections 2 and 3, a comprehensive overview of the growth of Chasm 1 and Halloween Crack is presented, based on data from a network of permanent GPS stations and strain stakes, extensive over-snow radar surveys, panchromatic Landsat-7/8 and Sentinel-2 satellite images, and novel methods using Sentinel-1A/B radar data. Some of the data is subsequently used in

Section 4 to configure an ice flow model of the ice shelf, and to perform a series of fracture propagation experiments that test our methods and provide a prediction for future propagation of both rifts. A discussion of the most important results in relation to the existing literature is provided in Section 5 and conclusions are given in Section 6.



## 2 Recent reactivation of Chasm 1

Chasm 1 is a 22-km long and, in places, up to 2-km wide rift structure that cuts through the middle of the BIS from the southwest to the northeast. The rift originated at the grounding line in the 1970s, where the ice shelf is only weakly connected to the continent, and was advected by the ice flow to its present-day location depicted in Figure 1. The formation of the rift

coincided with a period of accelerating ice flow, which started in the 1970s (Gudmundsson et al., 2017) and was likely related to wide-spread dynamical changes to the ice shelf triggered by a calving event to the north of the MIR (Thomas, 1973). After an initial phase of widening and growth, early satellite images showed that the location of the northernmost tip of Chasm 1 has remained unchanged with respect to the surrounding ice shelf since at least February 1980. Since the start of the Landsat-7 and 8 era in the early 2000s, the summer extent of Chasm 1 has been tracked more reliably and at regular intervals using all

available cloud-free panchromatic images, resulting in a time series of 35 manual outlines presented in Figure 2a. Up until late 2012, the rift showed little change to its overall shape and extent. However in November 2012, after more than 3 decades of inactivity, the tip of Chasm 1 started to propagate along a linear trajectory towards the MIR in the north, covering a total distance of about 7.3 km in 5 years.

### 2.1 Propagation of the Chasm 1 rift tip

In order to track temporal changes to the length of Chasm 1, the distance between the crack tip and a fixed reference point on the ice shelf has been calculated. The reference point is indicated by *A* in Figure 2a, and corresponds to a persistent surface feature that gets advected with the flow, known as a deflation hollow (Konovalov, 1964; Thomas, 1973). It is visible in all the satellite images used. The propagation of Chasm 1 is approximated by the straight-line distance between *A* and the tip of the rift as estimated from the outlines in Figure 2a. Results for this analysis are shown in Figure 2c and point towards a near-linear

growth between late 2012 and early 2017, with two episodes of significant slow-down in 2012 and 2016.

The accuracy of these results is limited by our inability to detect potential sub-surface propagation of the cracks, the variability in image contrast due to solar azimuth and zenith angles, and the resolution of the panchromatic images (15 m), which prevents the reliable detection of fine surface cracks less than a pixel wide. It is also impossible to track progress during the austral winter months (April - September) when visible images are unavailable. In order to address these issues and improve

the accuracy of the location of the rift tip, ground penetrating radar (GPR) measurements were acquired once every month from January 2016 to February 2017. Surveys were carried out using a radar unit with 400 MHz antenna from Geophysical Survey Systems, Inc., set to operate at a depth range of 30 m and tied to a dual frequency GPS to guarantee accurate geolocation of the radar traces. During each survey, a series of parallel lines was acquired covering the area around the rift tip, with a spatial separation of 100 m and directed approximately perpendicular to the propagation direction of the rift. An example of such a

set of survey lines for 24 February 2017 is shown by the white lines in Figure 2b.

The GPR data were detrended and a divergence compensation scheme was applied to enhance the intensity of the deeper layers, using the post-processing software package ReflexW from Sandmeier Geophysical Research. The processed data was used to detect narrow cracks less than 1 m wide, both at the surface and at depth, allowing to determine the exact extent of

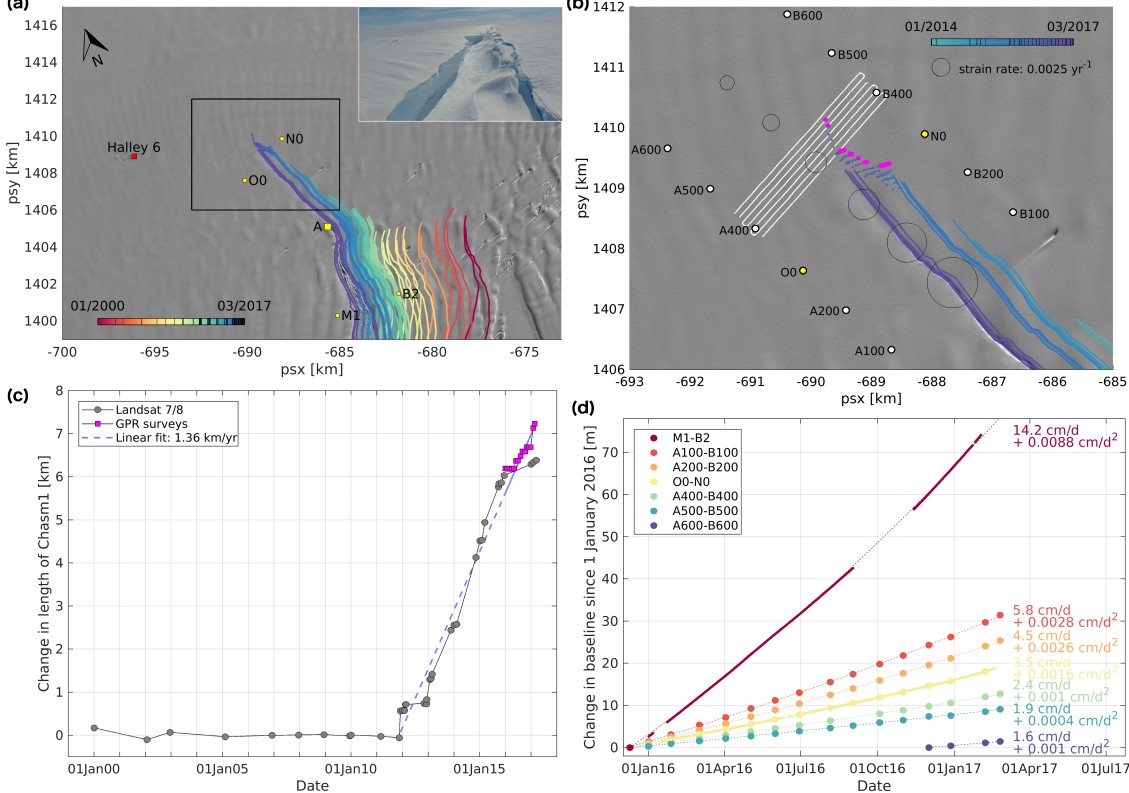

**Figure 2.** (a) Historical outlines of Chasm 1 from 6 January 2000 (red) to 15 March 2017 (purple), as the rift got advected by the ice flow. Outlines were obtained from a sequence of Landsat-7 and Landsat-8 panchromatic images, with acquisition dates as indicated by the black ticks in the colour bar. The background image is a subset of a Landsat-8 scene from 15 March 2017. The red dot indicates the location of Halley 6 research station, yellow dots are the locations of 4 permanent dual frequency GPS stations, and the black box outlines the extent of Figure b. The inset in the top right shows an aerial image of Chasm 1 taken in December 2015, looking from the reference point $A$ towards the crack tip. (b) Detailed overview of the area around the tip of Chasm 1, showing the local strain rate network (white dots), an example of the GPR survey lines (white lines), and progression of the crack tip as obtained from satellite (blue-to-purple outlines) and GPR data (blue-to-purple dots). Circles represent the widening of Chasm 1, with the radius of each circle scaled with the magnitude of the local strain rate. (c) Propagation of the tip of Chasm1 with respect to its historical location at the Gatekeeper prior to the reactivation in December 2012, based on Landsat-7/8 images (grey markers), and monthly GPR surveys (magenta markers). A linear fit through all datapoints shows an average propagation speed of 1.36 km/yr. (d) Baseline distance across Chasm 1 as a function of time, measured by two pairs of permanent GPS stations ($O0$-$N0$ and $M1$-$B2$ in (a)), and five pairs of strain rate stakes ($A_i$-$B_i$, with $i \in \{100, 200, 400, 500, 600\}$ in (b)). The least-squares quadratic fit is plotted as a dashed line.

Chasm 1 much more precisely, and at regular time intervals both in summer and throughout the winter. The blue-to-purple dots in Figure 2b represent all locations where cracks were detected within the vertical range of the GPR, with the colours indicating the acquisition date.



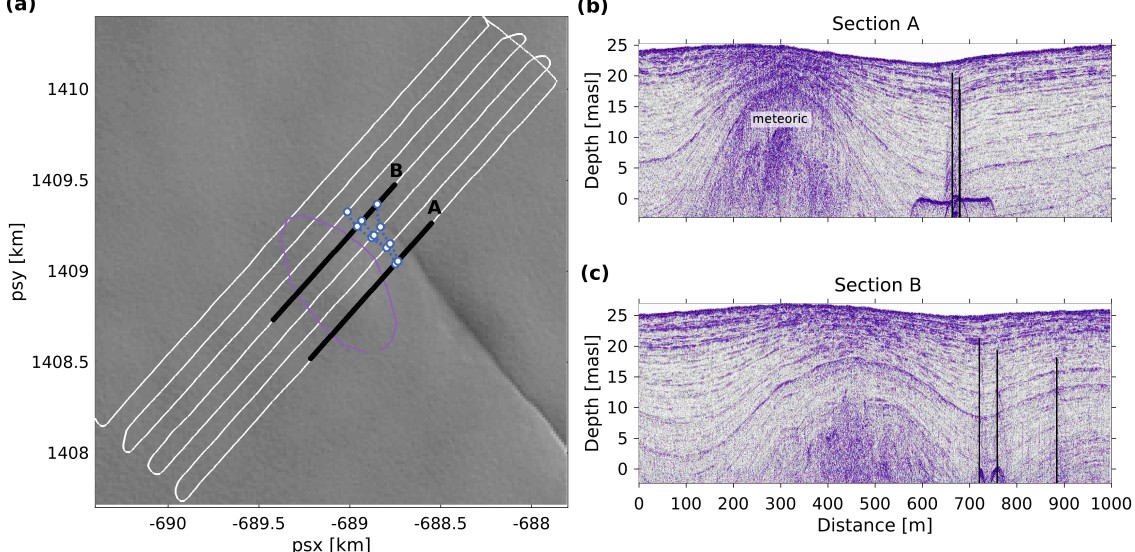

**Figure 3.** (a) GPS track of a GPR survey carried out on 4 May 2016 (white line), overlying a Landsat 8 image from 3 March 2016. Connected blue dots correspond to locations where a fracture has been observed in the GPR data. Black line segments A and B correspond to the location of the radar sections displayed in Figures b and c. The purple outline corresponds to the location of a large structure of meteoric ice embedded within the ice shelf (b) Radar section along line segment A in a. Vertical black lines indicate the location of fractures in the ice shelf. (c) Similar to b but for line segment B.

In general, the tip of Chasm 1 is not uniquely defined, but consists of a number of parallel branches at a spacing of several hundred meters, approximately aligned with the overall propagation direction. This non-simple structure of the rift tip is particularly evident between 1 January 2016 and 1 January 2017 during a phase of reduced rift tip propagation, and is illustrated in Figure 3 using data from a GPR survey carried out on 4 May 2016. Figure 3a shows the GPR track in white, overlying a

Landsat 8 image that predates the survey by about two months. Two sections are highlighted and the corresponding radar profiles are shown in Figure 3b and c, allowing the identification of two fractures along section A, and three fractures along section B, all of which are marked by a vertical black stripe. All branches follow the edge of a large structure within the ice, marked by "meteoric" in Figure 3b as it has been identified as a block of meteoric ice that originated at the grounding line and got embedded within the firn pack. Its spatial extend is marked by the purple outline in Figure 3a, although this is not a

unique feature and similar structures of variable size and orientation have been observed all across the ice shelf. It is very likely that such large-scale and wide-spread irregularities in ice properties are responsible for a significant dispersion of the crack tip growth. As the tip of Chasm 1 continued to propagate, some branches eventually became inactive, whereas others continued to grow. From January 2017 onwards, only one branch was apparent near the tip.

For each monthly GPR survey, the tip of the most advanced branch of Chasm 1 is highlighted in magenta in Figure 2b,

and the corresponding extent of the rift is represented by magenta markers in Figure 2c. A comparison between the GPR data



and satellite-derived results in Figure 2c shows that the latter systematically underestimate the extent of Chasm 1 by up to a kilometre. Not only are satellite data resolution limited, but GPR data also show that the tip of the rift tends to be further advanced at depth, and only becomes detectable at the surface at a later stage. In agreement with the satellite data, GPR measurements show a deceleration in the propagation between January 2016 and early 2017. However, a subsequent surge of

the crack tip in January and February 2017 was only captured by the GPR data, and satellite data alone would have led to the erroneous conclusion that Chasm 1 has stagnated until propagation once again became apparent.

## 2.2 Widening of Chasm 1

The gradual lengthening of Chasm 1 in the horizontal plane is known as a Mode I fracture caused by tensile stresses normal to the fracture plane. After fracture initialisation, such stresses tend to widen the crack at a rate which a-priori depends on the

far-field stresses and on material properties local to the fracture tip. The precise nature of this relationship cannot be determined due to a lack of observational data, however for Chasm 1 a network of strain stakes and permanent GPS stations was installed to monitor crack widening rates. The network is shown in Figure 2b, and consists of 5 stakes (white markers) at 1 km separation on each side of Chasm 1, named $A100$-$A600$ on the west and $B100$-$B600$ on the east side. The relative location of all stakes was measured once a month between December 2015 and February 2017, using a GPS base station at $A600$ and roving stations

at all other stakes. In addition, two permanent GPS stations were installed at $O0$ and $N0$ (yellow markers in Figure 2a), which have provided a daily measurement of their location since January 2016. A further pair of permanent GPS stations, named $M1$ and $B2$ in Figure 2a, was installed in May 2015, and has provided additional baseline measurements across Chasm 1 further towards the south and away from the tip. By combing these data, the widening of Chasm 1 has been measured along 7 different baselines, some across the historical extent of the crack ($M1$-$B2$), others across the recently formed branch ($A100$-

$B100$, $A200$-$B200$, $O0$-$N0$, $A400$-$B400$), and some ahead of the current crack tip ($A500$-$B500$, $A600$-$B600$). All data were post-processed using the advanced baseline analysis tool within the Infinity software package from Leica Geosystems, and a summary of the results from 1 December 2016 until 27 February 2017 is shown in Figure 2d.

All baselines that span across Chasm 1 get longer over time, ranging from less than 2 cm/day between $A600$ and $B600$, to 14.2 cm/day between $M1$ and $B2$. The rate of opening per unit distance along each baseline, or strain rate, is represented

by the circles in Figure 2b, with values ranging from 0.002 yr$^{-1}$ along $A600$-$B600$ to 0.007 yr$^{-1}$ along $A100$-$B100$. Part of the measured strain can be attributed to the background spreading of the ice shelf, as is apparent along $A500$-$B500$ and $A600$-$B600$ where the crack has not yet propagated, whereas the remainder of the extension is due to the widening of the crack. The strain rates show an increasing trend away from the crack tip, and have a non-negligible second order component in time, which indicates that Chasm 1 is widening at an accelerating rate. Unlike the spatial propagation of the tip location, the

widening of Chasm 1 happens evenly and without significant deviations from the trend line. For example, the transition from a period of slow crack propagation to a rapid change in the location of the crack tip in early 2017 (Figure 2c) is not obviously reflected in the widening rates. This suggests that the widening is dominated by the slowly varying far-field stresses, and is uncorrelated to the rift tip propagation, which is sensitive to spatially variable properties of the ice shelf local to the tip, such as the ice viscosity and ice thickness. Similar observations have previously been made by (Joughin and MacAyeal, 2005) for



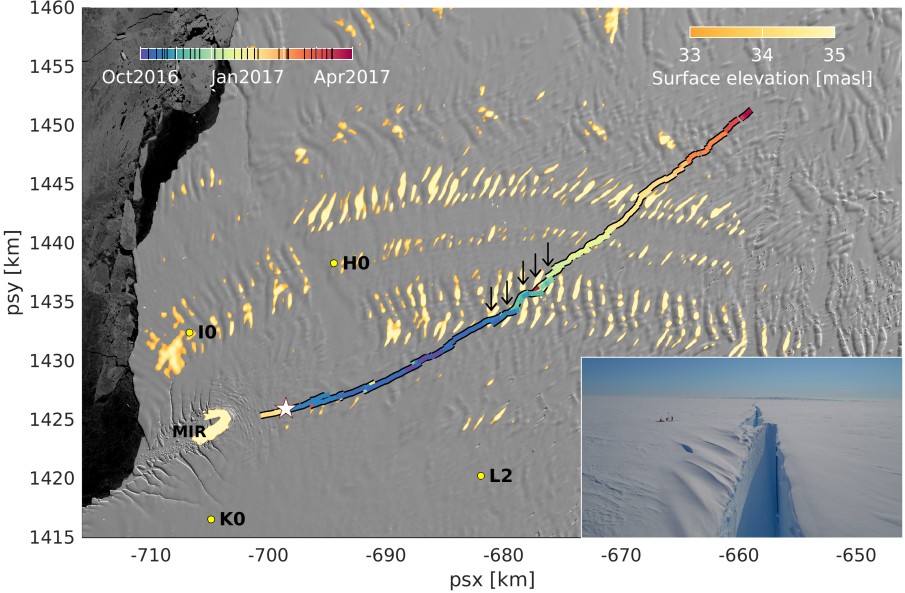

**Figure 4.** Extent of Halloween Crack (HC) based on a time series of manual outlines from Landsat-8 and Sentinel-2 images; the corresponding dates are indicated by black ticks in the colorbar. The background image is a Landsat-8 scene from 15 March 2017, yellow dots correspond to the location of 4 permanent GPS stations (K0, I0, H0 and L2), and the inset in the lower right shows an aerial image of the HC taken in January 2017 from a location indicated by the white star and looking towards the MIR. Orange-to-yellow colours highlight areas with a surface elevation above 33 m, and arrows point towards locations where the HC cuts through a band of thicker ice.

two rifts that led to the calving of tabular icebergs from the Ross Ice Shelf, Antarctica, and we will come back to this result in section 5.

## 3   Halloween Crack formation and propagation

On 31 October 2016, airborne observations revealed a new rift in the BIS, extending from the MIR towards the east over
5   a distance of about 15 km. The earliest evidence for this fracture, henceforth named *Halloween Crack* or *HC*, dates back to a Landsat-8 image from 11 October 2016, whereas an earlier image from 29 September 2016 only revealed a number of disconnected segments extending over a combined length of 2 km. A later outline from 15 March 2017 is shown in Figure 1, and illustrates the extent of the HC at the onset of austral winter. By that time, the crack had grown an additional 35 km in length, and cut across the BIS along a west-east trajectory, roughly parallel to the grounding line, and at a 90 degree angle to
10   Chasm 1. This rapid growth has been tracked at frequent intervals using Landsat-8 and Sentinel-2 panchromatic images, and a total of 30 manual outlines were acquired between 1 January 2016 and 15 March 2017. The resulting time series is shown in Figure 4 with colours indicating the corresponding acquisition date.





### 3.1 Propagation of the tip of Halloween Crack

The HC originated about 15 km east of the MIR, and its tip propagated rapidly in opposite directions. From November 2016 onwards, growth stagnated in the west as the rift approached the MIR, but continued at an unabated rate towards the east. When plotted as a function of time (see blue dots in Figure 5a), the total length of the HC increased by 34.2 km over 158 days, or

an average of 216 m/day, subject to observational errors related to our inability to detect subsurface or narrow surface cracks using visible satellite images, as discussed in Section 2.1. Discarding such uncertainties, it is clear that the propagation rate has been highly variable in time, with very little, if any, propagation in early November 2016 and periods during January, February and March 2017, to more than 600 m/day during intervening periods.

This variability has also been observed for Chasm 1, and is at least partly related to the heterogeneous structure of the

ice shelf. As the tip of the rift propagates, it encounters areas with higher (lower) fracture toughness, which requires a higher (lower) energy to break through, and leads to a slow-down (speed-up) of the propagation. In the case of the HC, a period of slow propagation from mid-October to mid-November 2016 occurred when the crack tip encountered successive bands of thicker meteoric ice, apparent in the surface topography in Figure 4 and indicated by black arrows. These blocks of ice originated at the grounding line, where the grounded ice calved at regular intervals leaving spaces between icebergs that were gradually

filled by sea ice, marine ice and surface accumulation. As the heterogeneities and the direction of propagation were misaligned by about 70 degrees, the HC changed direction as it followed a staircase-like trajectory with successive phases of propagation along and perpendicular to the icebergs. This caused temporary changes in its alignment with respect to the large-scale stress field, which might explain part of the slow-down. Other studies have suggested that the presence of marine ice can suppress the speed at which rifts propagate, and several ice shelves such as Larsen C and Filchner-Ronne Ice Shelf contain rifts that

terminate in marine ice-rich suture zones (Glasser et al., 2009; Hulbe et al., 2010; McGrath et al., 2012; Jansen et al., 2013; Borstad et al., 2017). However, very few studies have investigated the relationship between rift tip propagation and fracture toughness of ice directly, see for example Rist et al. (2002), and the dynamics of rift tip propagation remains subject to large uncertainties.

### 3.2 Widening of Halloween Crack

The sudden fracture of the ice shelf and the abrupt formation of the HC suggests a period of critical crack growth, which typically occurs in materials that are subjected to high tensile stresses, or stresses that are applied for long enough such that initial cracks grow to a critical length, after which catastrophic failure happens (Lawn, 1993; Rist et al., 2002). Satellite images from 2014 and 2015 show early indications of small segmented fractures at the surface in the area where the HC subsequently originated, and it is feasible that in October 2016, these segments reached a critical length, which led to the rapid formation

of the HC. During this short phase of critical growth, the propagation speed might have reached a considerable fraction of the speed of sound, and continued until changes in the remote stress field stabilised the crack and a quasi-static regime with slower growth rates was reestablished.





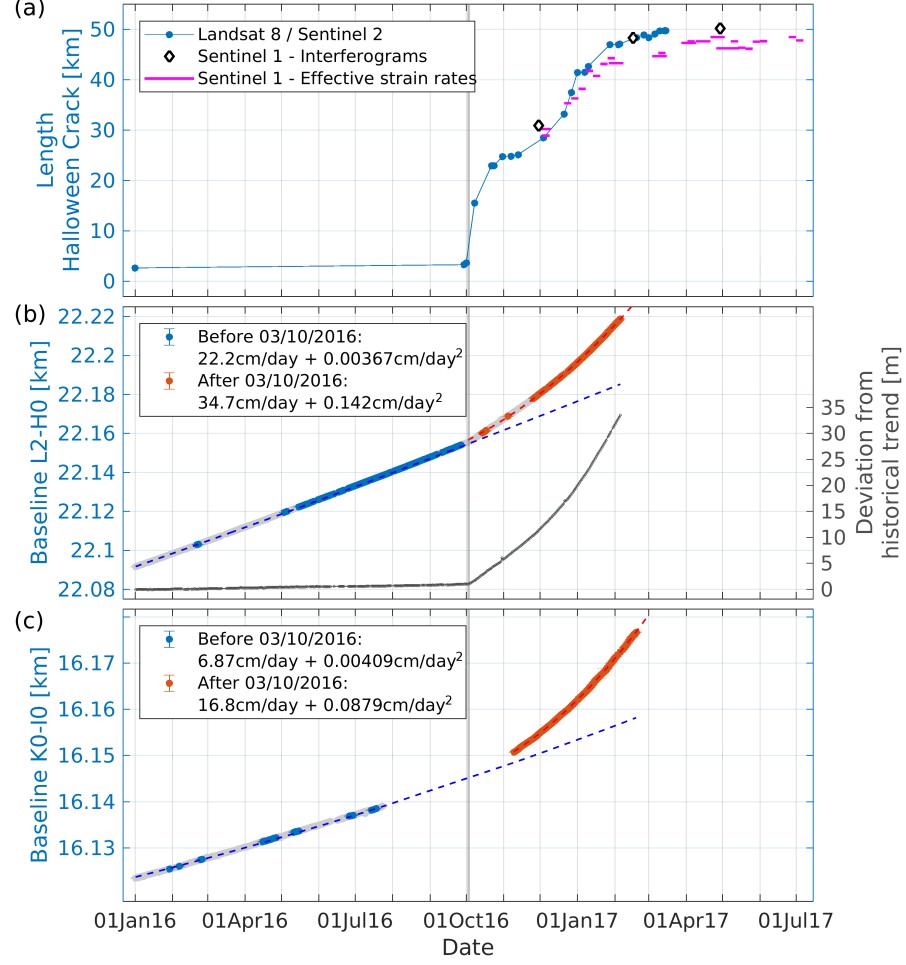

**Figure 5.** (a) The blue curve represents the length of Halloween Crack (HC) as a function of time based on a sequence of outlines from visible Landsat-8 and Sentinel-2 images. Black diamonds indicate the length of the HC based on manual tracking of the rift tip in a sequence of interferograms (see Figure 6). Magenta lines correspond to the length of the HC defined as the extent of the 0.025 $yr^{-1}$ contour of the effective strain rate, derived from Sentinel-1A/B velocity fields (see Figure 7b). (b) Left axis: widening of the HC along baseline L2-H0 in Figure 4; right axis (black line): deviation from the historical trend prior to the formation of the HC on 3 October 2016. (c) Same as (b) but for GPS stations K0 and I0. The grey vertical line in each image highlights the period around 3 October 2016, when the HC was formed.

Once failure occurred, the ice adjacent to the free surfaces of the crack became fully unloaded, and the strain was entirely converted into opening of the crack. The transition from a loaded to an unloaded state of the ice shelf was measured using a network of 4 permanent GPS stations positioned on both sides of the HC trajectory. As shown in Figure 4, stations $K0$ and $L2$ were located south of the HC, and stations $I0$ and $H0$ were positioned towards the north, forming two independent baselines

5 across the rift: $K0$-$I0$ and $L2$-$H0$.





Daily values for the length of each baseline were obtained using the baseline processing toolbox in Leica Infinity, and results are presented in Figure 5b ($L2$-$H0$) and Figure 5c ($K0$-$I0$). Phase-fixed solutions with a precision of about 1 mm were obtained for 45% of the measurements despite the length of the baselines (16 km and 22 km), and are highlighted in blue and red in Figures 5b and c. The remaining float solutions have a precision on the order of 5 cm and are colored in grey. Although

all stations were installed prior to 2013, only data after 1 January 2016 was used in this study, providing 10 months of coverage leading up to the formation of the HC in early October 2016. Note that a failure at $I0$ caused a gap in the $K0$-$I0$ baseline between August and November 2016, and both time series were ended in February 2017, when the most recent data was collected before the start of austral winter.

Both baselines across the HC showed a qualitatively similar behaviour over the measurement period, with a close-to-linear

extension rate up until 3 October 2016 (dashed blue line in Figures 5b and c), followed by a rapid deviation from the historical trend line and a significant acceleration. The kink on 3 October 2016, which is more obvious from the anomaly plotted on the right axis in Figure 5b (black line), provides a precise indication of when the HC formed. On that date, spreading rates along $L2$-$H0$ increased near-instantaneously from 22.2 cm/day to 34.7 cm/day with an acceleration of 0.142 cm/day$^2$, and extension along $K0$-$I0$ more than doubled from 6.87 cm/day to 16.8 cm/day. The sharp increase in extension rate across the

HC on 3 October can be fully attributed to the widening of the HC. When compared to variations in the length of the HC after 3 October 2016 (Figure 5a), there is no significant correlation between crack tip propagation and rift widening, which hints at different primary controls for both processes, and confirms earlier findings for Chasm 1.

In order to identify the source for the increase in baseline extension after 3 October 2016, absolute locations of the GPS stations were obtained from a precise point positioning analysis using the Bernese GNSS software. Results indicate that the

signal is caused by an acceleration of the surface velocities at $I0$ and $H0$ to the north of the HC, following the physical separation of the northern and southern parts of the ice shelf. This signal has subsequently been confirmed by satellite derived surface velocities, which show a clear discontinuity in surface velocities across the rift with higher velocities towards the north across its entire length. An example of such a velocity field for May 2017 is shown in Figure 7a.

## 3.3 Tracking of Halloween Crack using SAR data

Due to the fast propagation of the HC and the complexity of the terrain, it is unfeasible to carry out ground-based radar measurements to accurately track the location of the crack tip. In addition, satellite coverage of the ice shelf in the visible spectrum is lost during the austral winter months, so alternative methods need to be developed to provide year-round coverage of the propagation. Data from the Sentinel-1A/B satellites provides a viable alternative, as they carry an active C-band synthetic aperture radar (SAR) instrument that is not affected by cloud cover or the absence of visible light.

A well-known method to study fracture propagation using SAR data exploits the phase difference between two images taken several days apart, and is known as Interferometric SAR or InSAR. This method is routinely used in earth surface deformation monitoring, see e.g. Bürgmann et al. (2000) for a review, ice flow monitoring, see e.g. Goldstein et al. (1993) for one of the earliest examples, or grounding line tracking. Its application to ice shelf rift deformation has been limited, see e.g. Larour et al.



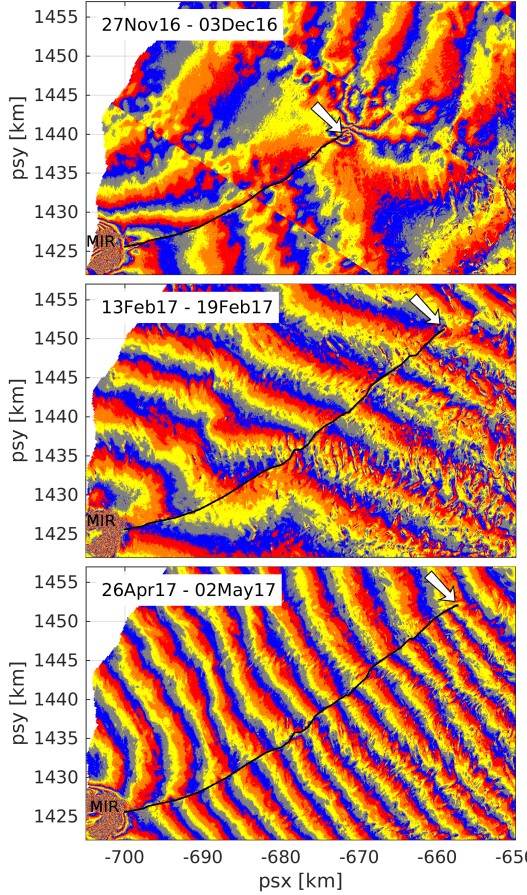

**Figure 6.** A selection of interferograms obtained about two months apart, showing the progress of Halloween Crack as it propagated from the McDonald Ice Rumples (MIR) towards the east. White arrows point towards the eastern tip of the crack.

(2004a), although there has been a renewed interest in the method due to the improved spatial and temporal coverage of SAR data for Antarctica's ice shelves.

Discontinuities in the differential phase (or interferogram) typically correspond to areas of anomalously high surface deformation, which can be traced manually and associated with active fracture zones. The technique is demonstrated in Figure 6, which shows three interferograms for the area around HC separated by about 2 months. In each image a discontinuity in the interferometric fringes marks the trajectory of the crack, as it propagates from the MIR towards the east. Black diamond markers in Figure 5a specify the length of each outline, and a good agreement is found between SAR-derived estimates of the crack length and earlier results obtained from visible satellite images.

Despite the success of this technique, interferograms are often difficult to interpret, especially in places with a complex surface topography such as the BIS, and phase discontinuities caused by artefacts in the data are easily mistaken for physical





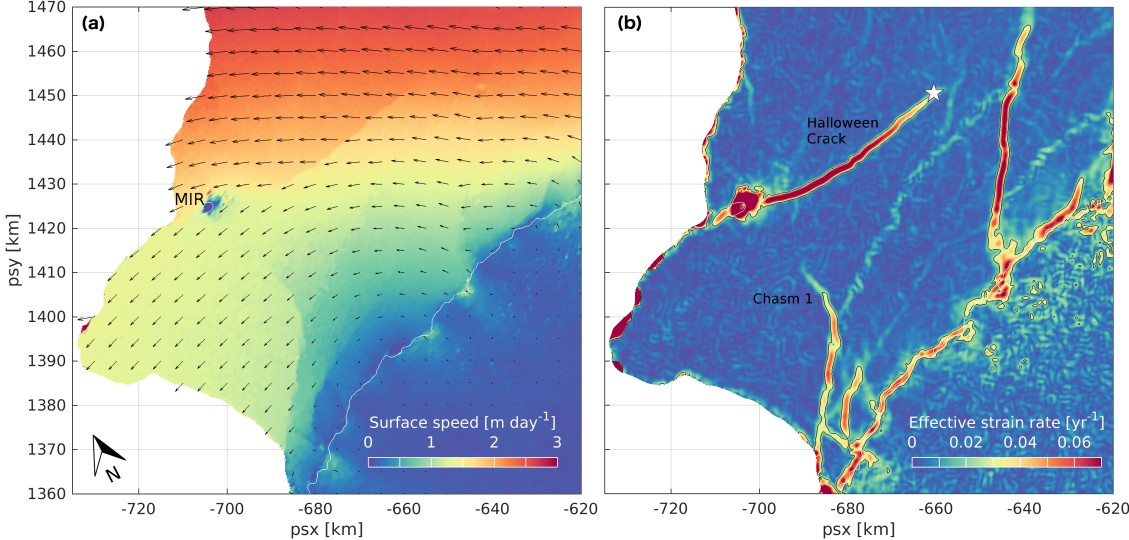

**Figure 7.** (a) Surface velocity (arrows) and speed (colours), obtained from a Sentinel-1 image pair (14 and 20 May 2017) using an iterative offset tracking method. (b) Effective strain rates derived from the velocity field in (a), using a quadratic regression method as detailed in the main text. The black line corresponds to the $\epsilon_e = 0.025$ yr$^{-1}$ contour, and the white star indicates the easternmost extent of the HC as outlined by the contour.

signals. The method is also less appropriate for slowly propagating rifts with surface deformations that are too small to detect over the 6-day repeat cycle. To avoid these issues, a more quantitative and robust approach to monitor fracture deformation is explored, based on SAR-derived velocity fields and the corresponding horizontal strain rates. As the dual satellite configuration of Sentinel-1A/B provides complete coverage of the BIS every 6 days, it is ideally suited for the frequent recovery of surface

velocities using an iterative offset tracking method developed by Nagler et al. (2015). As an example, Figure 7a shows the horizontal velocity field derived from two Sentinel-1 SAR images taken on 14 and 20 May 2017. The horizontal strain rate components $\dot{\epsilon}_{xx}$, $\dot{\epsilon}_{xy}$, and $\dot{\epsilon}_{yy}$ were calculated using a local quadratic regression of the velocity field over a 2 km x 2 km square, and used to obtain the effective strain rate $\dot{\epsilon}_e = \sqrt{\left(\dot{\epsilon}_{xx}^2 + 2\dot{\epsilon}_{xy}^2 + \dot{\epsilon}_{yy}^2\right)/2}$ shown in Figure 7b. Average values across the ice shelf are less than 0.01 yr$^{-1}$, whereas higher-than-average strain rates up to 0.1 yr$^{-1}$ are found in the vicinity of the MIR, and along

the HC and Chasm 1. Other areas with high strain are the grounding line and a shear margin, which extends from the grounding line at around $x = -645$ km towards the northeast.

In order to define an objective measure for the extent of the HC based on strain rates, the $\dot{\epsilon}_e = 0.025$ yr$^{-1}$ contour is used as an approximate outline of the crack, and the easternmost extent of the contour is used as a proxy for the location of the crack tip. From this location, indicated by the white star in Figure 7b, the length of the HC can be calculated. This process removes

biases related to the manual interpretation of images, and has been applied to 22 velocity maps and corresponding effective strain rates, acquired by Sentinel-1A/B between November 2016 and July 2017. For each acquisition, the $\dot{\epsilon}_e = 0.025$ yr$^{-1}$



contour was identified, and used to track the extent of the HC. The results are shown by the magenta lines in Figure 5a, where the horizontal limits of each line indicate the acquisition dates of the SAR image pair used to derive the velocity field.

The precision of our method to estimate the length of the rifts can be determined by comparing results for overlapping periods, and shows a spread of around 2 km. A comparison to manual tracking from visible images (Landsat-8, Sentinel-2)

shows that the two independent measurements are generally in good agreement, although there is a bias towards smaller values for the SAR-derived values. This is in part due to a systematic underestimation of the strain rates at the crack tip due to smoothing by the quadratic regression method, and in part because the $\dot{\epsilon}_e = 0.025 \ \mathrm{yr}^{-1}$ contour fails to capture the lower strain rates near the very tip of the crack. However, the record is internally consistent and, importantly, presents a reliable extension beyond previous summer measurements. It shows a sustained reduction in propagation speed of the HC since February 2017,

as the tip covered about 5 km or 32 m/day between early February 2017 and mid July 2017, compared to 366 m/day during the first four months of propagation.

## 4   Model predictions of rift propagation direction

As the satellite-derived effective strain rates ahead of the tip of the HC and Chasm 1 in Figure 7b are not significantly different from the average far-field strain rates across the ice shelf, they cannot be used as a direct indicator for future crack propagation.

Moreover, effective strain rates derived from a Sentinel-1 velocity map in June 2015 (not shown) do not contain any evidence for the nascent formation of the HC, indicating that they cannot be used with confidence to predict when and where cracks will form in the ice shelf. However, information in the *directionality* of the stress field can be used to estimate the trajectory of propagation, once a crack has formed. Such methods range from sophisticated modern fracture mechanics including linear elasticity theory (Rist et al., 2002), to simple criteria based on the direction of maximum tensile stress (Ingraffea, 1987). As

the first principle stress across the ice shelf is tensile, it indicates the direction perpendicular to which fractures are most likely to propagate. Here we present a simple approach to simulate the propagation direction of existing fractures based on the latter method, and test our results against the observed trajectory of the HC.

### 4.1   Model setup and fracture propagation algorithm

In order to predict the propagation direction of existing fractures in the BIS based on the observed stresses, surface velocities

from Sentinel-2 and ice thickness data were assimilated in the SSA flow model Úa (Gudmundsson et al., 2012). The boundaries of the model domain were chosen to coincide with the grounding line and icefront of the BIS, and extended a further 150 km towards the east to include the neighbouring Stancomb Wills Glacier Tongue, in order to fully capture the spatial variability in strain rates, and to allow a sufficiently large domain to propagate the existing rifts. A triangular mesh with linear elements was used, with the distance between nodes varying from 250 m for the BIS, to 1 km elsewhere. The flow at the grounding line was

prescribed by a Dirichlet condition, and Chasm 1 was represented as an ice free area, i.e. a gap in the model domain.

Ice thickness values for the BIS were derived from a high resolution (3 m) WorldView surface DEM acquired between 2012 and 2014, and values were binned onto a coarser 1 km x 1 km grid. Elevations above the ellipsoid were translated into





elevations above sea level using a tidal correction (Gudmundsson et al., 2017) and a EIGEN-6C geoid correction, and converted into ice thickness values assuming floatation and a 2-layer density model with a constant 30 m firn column (750 kg/m$^3$) overlaying pure ice (920 kg/m$^3$). For the Stancomb Wills Glacier Tongue a subset of the Antarctica-wide iceshelf thickness dataset by Chuter and Bamber (2015) was used.

The mismatch between observed and modelled surface velocities was minimised through an iterative optimisation (or inversion) of the rate factor in Glen's flow law with exponent $n = 3$, and the resulting stress field was used to calculate the fracture trajectories. Each crack was seeded at a predefined location in the model, informed by the observed location of an initial fracture or the tip of an existing rift in the ice shelf. Subsequently, the local direction of maximum tensile stress was identified from the modelled stresses, and the crack was propagated perpendicular to this direction over a fixed distance. The

step size was chosen to be on the order of the mesh resolution, i.e. 250 m, although results were shown to not critically depend on this choice. The process was iterated for a number of steps, or until a certain crack length was reached. It should be noted that the algorithm is diagnostic and does not contain a time-component, nor does it include any dynamic feedback between the formation of the rift and the background stress field.

    Two separate inversions were performed with Úa, one for surface velocities in late June 2015, prior to the formation of the

HC, and a second inversion for mid March 2017, after the formation of the HC. The first inversion was used to hindcast the propagation trajectory of the HC as a way to validate the model and methods, the second inversion was used to predict the future direction of propagation for Chasm 1 and the HC using a more recent stress field. Results are discussed in Section 4.2 and Section 4.3 respectively.

## 4.2   Model validation

For validation purposes, the first experiment was designed to hindcast the observed trajectory of the HC based on the state of stress in the ice shelf shortly before fracture formation in October 2016. For this purpose, a surface velocity field obtained from a Sentinel-1 image pair on 18 and 30 June 2015 (Nagler et al., 2015) was assimilated in Úa. The corresponding tensile stress orientations were used to propagate a crack in opposite directions from the location indicated by the magenta circle in Figure 8a, which corresponds to the location where the HC was first observed. The resulting trajectory is shown by the magenta

line in Figure 8a and follows the observed extent of the HC (in black) to within 1 km along its entire length. This excellent agreement between model results and observations underlines the validity of the simple fracture propagation algorithm, and provides confidence in the predictive skills of the method.

    The success of the algorithm relies on an accurate representation of the stresses in the ice shelf, which largely depends on the accuracy of the inversion procedure in Úa. The mean residual between observed and modelled surface speeds for the BIS was

found to be $1.8 \pm 9.6$ m/yr or $0.2 \pm 2.1\%$, which is a significant improvement over De Rydt et al. (2015) and Gudmundsson et al. (2017) due to recent improvements in the inversion procedure in Úa, and Larour et al. (2014), who found a best match for the BIS of around 10%. The principle stress rosettes in Figure 8a (red arrows are compressive, blue arrows are extensive) display a radially symmetric pattern around the MIR, with compressive stresses at oblique angles to the flow in the upstream direction, and compensating tensile stresses perpendicular to the flow. This pattern is generated by the point-interaction between the



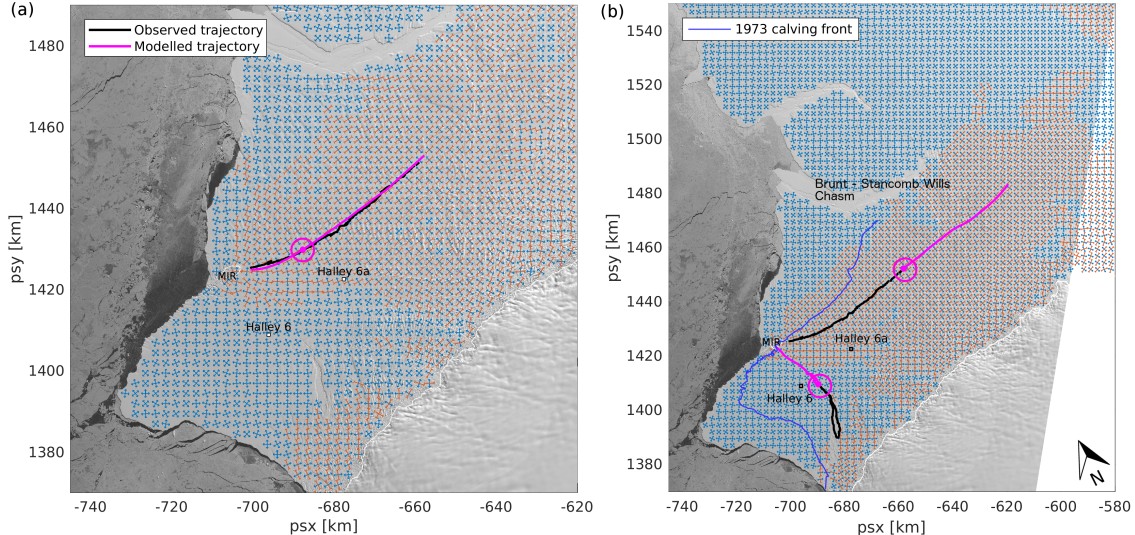

**Figure 8.** (a) Modelled principle stress components (red are compressive, blue are extensive) in June 2015. The modelled trajectory of Halloween Crack based on the direction of maximal tensile stress is shown in magenta, the observed trajectory from 15 March 2017 is plotted in black. (b) Modelled principle stress components in March 2017 and predicted future trajectories of Chasm 1 and Halloween Crack. The purple line corresponds to the 1973 calving front.

ice shelf and the bedrock at the MIR, and has resulted in the radially-outward growth of the HC, perpendicular to the tensile stresses.

## 4.3 Future fracture propagation

The second experiment was designed to propagate Chasm 1 and the HC beyond their current-day extent, based on the stress distribution in the ice shelf after the formation of the HC. The model domain was adjusted for a minor advance of the ice front between 2015 and 2017, and updated outlines of Chasm 1 and the HC for 15 March 2017 were introduced as gaps in the mesh. Surface velocities in Úa were optimized to match observed velocities based on a pair of Sentinel-1 images acquired on 9 and 15 March 2017, with a final residual between observed and modelled surface speeds of $-2.4 \pm 13.5$ m/yr or $-0.2 \pm 2.6\%$.

The model domain was seeded with initial fractures at the tips of the HC and Chasm 1, indicated by the magenta circles in Figure 8b. The rifts were propagated perpendicular to the direction of maximum tensile stress, up to the edge of the model domain for Chasm 1 and for a total length of 50 km for the HC, as shown by the magenta lines in Figure 8b. According to these trajectories, the HC is expected to continue along its current west-east trajectory towards the Stancomb Wills Glacier Tongue, cutting across a region with uniform tensile stresses that are aligned north-south perpendicular to the grounding line, and second principle stresses that are compressive throughout. The estimated trajectory therefore cuts deep into an area of buttressed ice, which could have important implications for the structural integrity of the ice shelf (Doake et al., 1997; Fürst



et al., 2016). Note that the estimated trajectory was ended after 50 km because of possible interactions with the active Brunt - Stancomb Wills Chasm (see Figure 8b and Anderson et al. (2014)), which could affect the growth of the HC in uncertain ways.

Furthermore, based on the current stress configuration, the model predicts a continued propagation of Chasm 1 towards the MIR, as shown in Figure 8b. The path passes through a region with extensive stresses in both principal directions before cutting

into the compressive stress region south of the MIR, which could give rise to a possible further reduction in buttressed ice.

## 5    Discussion

Rift propagation in ice shelves is a timely subject and an important process that is often ignored in studies of future ice loss from the Antarctic continent, due to the large uncertainties associated with the prediction of rift formation and propagation, and the lack of an adequate theoretical description or "calving law". In this work, a number of observations have been made,

which confirm or refute findings in earlier work, and highlight the complexity of the subject.

Widening rates of both Chasm 1 and the HC accelerate with rift extension, consistent with previous observations and modelling work by Larour et al. (2004a, b) and Joughin and MacAyeal (2005). Based on the unique timeseries of baseline measurements across Chasm 1 and the HC, presented in Figures 2d, 4b and 4c, we postulate that over the measurement period, widening rates follow an approximately linear relationship in time:

$$\frac{\partial w(x,t)}{\partial t} = w_0(x) + w_1(x)t + \varepsilon(x,t), \tag{1}$$

where $w(x,t)$ is the edge-to-edge width of the rift at a location $x \le x_{\text{tip}}(t)$ along the rift. The origin $x = 0$ corresponds to the root of the rift, *furthest away* from the tip. The small residue term, $\varepsilon(x,t)$, incorporates measurement errors as well as second-order effects that will be discussed below.

The functions $w_0(x)$ and $w_1(x)$ depend on the distance along the rift, and have been measured along five different baselines

across Chasm 1, i.e. $M1$-$B2$, $A_{100}$-$B_{100}$, $A_{200}$-$B_{200}$, $O0$-$N0$ and $A_{400}$-$B_{400}$ in Figure 2d, for a period from January 2016 until January 2017. A regression analysis shows that $w_0$ and $w_1$ are linearly dependent on $x$ with $R^2$ values of 0.997 and 0.948 respectively, and there is no discernible time-dependence over the measurement period of one year. In summary, the observational data provides strong evidence for an empirical relationship of the form

$$\frac{\partial w(x,t)}{\partial t} = w_{00} + w_{01}x + w_{10}t + w_{11}xt + \varepsilon(x,t), \tag{2}$$

with constant parameters $w_{ij}$.

A preliminary analysis of the second-order effects, which have been absorbed into $\varepsilon(x,t)$ in equation 2, reveals a direct relationship between rift widening and the propagation speed of the rift tip, and a clear response to tidal forcing. However, the relative amplitude of such effects, i.e. $\left| \frac{\partial w}{\partial t} - \varepsilon \right| / \frac{\partial w}{\partial t}$, is less than $4\%$ and therefore negligible. In particular, the propagation speed of the rift tip, which happens in alternating phases of slow and fast movement in response to spatially variable properties

of the ice shelf, has no important influence on the widening rates, and we find no evidence for a stick-slip mechanism of tip propagation as proposed by Larour et al. (2004a). A more detailed analysis of the second-order variability will be presented



elsewhere, but based on the data presented here, we conclude that rift widening is a slowly accelerating process without any clear periodicity, driven by the large-scale distribution of stresses in the ice shelf. Moreover, in contrast to observations by Fricker et al. (2005), the dynamics of Chasm 1 shows no clear seasonal variability superimposed on the multi-year linear trend, and it is unclear if Chasm 1 and the HC are affected by ice melange that fills the rifts, a mechanism that was put forward by

Larour et al. (2004b). During austral summer, open water has been regularly observed at the bottom of both rifts, suggesting that widening rates are too fast for the formation of sea ice to close any potential leads and have any net effect on the stresses.

Based on the tensile stress distribution in the ice shelf, we were able to predict the path of the HC with remarkable accuracy, and future trajectories have been put forward for testing against new observations. In order to achieve such accuracy, observations were assimilated into an ice flow model, and trajectories were calculated from the modelled stresses. Although principle

strain rate directions from remote sensing velocity data could be used as an alternative to estimate the fracture propagation direction, such datasets typically suffer from a low signal-to-noise ratio, and the model effectively acts as a filter to reduce the noise. The suggested propagation algorithm makes use of the direction of maximum tensile stress, and is only applicable to existing rifts with known tip location. Observations have shown that the stress magnitudes are not a good indicator for the formation of new rifts, or for predicting future propagation.

The future shape of the ice shelf follows from the projected fracture trajectories, and the future of the BIS is most critically dependent on the amount of ice that is lost directly at the MIR, as it controls the reduction in buttressing and corresponding changes in stress over the rest of the ice shelf. During a previous calving event that took place between 1968 and 1971, a rift formed in a location similar to the HC, as shown in Figure 8b by the outline of the icefront derived from a Landsat-1 image on 1 January 1973, shortly after the calving event. At that time, the ice shelf lost a considerable amount of ice at the MIR, and

doubled its speed in response (Simmons and Rouse, 1984a), although the main part of the ice shelf to the south of the MIR remained intact. This scenario is likely to repeat itself as the HC develops into an iceberg, and although the ice front will be substantially further inshore compared to 1973, the amount of ice removed at the MIR could be comparable.

However, despite the similarities to the 1970s event, the present-day circumstances on the BIS are unprecedented because of the recent re-activation and uncertain interplay with Chasm 1. The potential for additional ice loss at the MIR and a further

reduction in buttressing due to the propagation of Chasm 1 is evident from the predicted trajectory in Figure 8b. This could greatly affect the dynamics and structural integrity of the remainder of the ice shelf, and only when the ice shelf maintains contact with the bedrock at the MIR, a re-advance and return to fully buttressed conditions can be expected in the future.

## 6   Conclusions

We have presented a comprehensive collection of ground-based and remote sensing data, providing a detailed overview of

glaciological changes on the Brunt Ice Shelf over the last decade. In particular, the reactivation of Chasm 1 in December 2012 and the formation of the Halloween Crack in October 2016 have been documented in great detail, and have provided an unprecedented view on the dynamics of such rifts, commonly found in Antarctica. We presented the acquisition of ground-based radar data with high temporal and spatial resolution in the area of the tip of Chasm 1, showing a complex structure of





the rift at its tip, which is invisible on satellite data. Over the measurement period, both Chasm 1 and the Halloween Crack have been widening at a slowly accelerating rate, and have propagated deep into the interior of the ice shelf during successive episodes of faster and slower growth, controlled by the heterogeneous structure of the ice shelf. Through the assimilation of observed ice thickness and surface velocity data into a shallow-ice flow model, and the use of a simple fracture propagation

criterion based on the stress distribution in the ice shelf, estimates were made for the future propagation direction of both rifts. Results show a conceivable future loss of buttressed ice in contact with the bedrock at the McDonald Ice Rumples, the only pinning point on the ice shelf, which could further contribute to the ongoing dynamic changes. An intensive monitoring programme has been put in place to track such changes, both in austral summer and winter, using GPS stations, ground-based radar measurements, and frequent satellite products from Landsat-8 and Sentinel-1/2. These data will continue to be used in

modelling activities that aim to enhance our understanding of ice shelf buttressing, rift propagation and calving – processes that are key to the future mass balance of this and many other ice shelves around Antarctica.

*Competing interests.* The authors declare no competing interests.

*Acknowledgements.* Processing of Sentinel-1 ice velocity maps by ENVEO was supported by the Austrian Space Applications Programme/Austrian Research Promotion Agency (FFG) and the ESA Antarctic ice sheet CCI project and with contributions by Markus Hetzenecker and Joanna

Ossowska, both at ENVEO. Copernicus Sentinel-1 data were made available through the ESA Sentinel Scientific Data Hub. J.D.R., G.H.G. and E.C.K. were partly supported as part of the *Polar Science for Planet Earth* funding from the Natural Environment Research Council (NERC) to the British Antarctic Survey.





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
