# Peer review of "Recent rift formation and impact on the structural integrity of the Brunt Ice Shelf, East Antarctica"

_The Cryosphere, 2017_

## Referee Comment (RC1) · Anonymous Referee #1 · 17 Oct 2017

Review of "Recent rift formation and impact on the structural integrity of the Brunt Ice Shelf, East Antarctica" by De Rydt et al.

This manuscript reports on detailed observations and model analysis of two rifts in the Brunt Ice Shelf. These rifts may threaten the structural integrity of the shelf, and at the very least have caused quite some concern for the fate of the Halley Research Station on the shelf. An impressive array of observations have been collected on the behaviour of these rifts, and the observations alone reveal quite a lot about the nature of rift propagation. A numerical model is applied to determine the orientation of principal stresses in the shelf, and a clever heuristic algorithm for manually modeling the trajectories of the cracks is applied. This heuristic proves successful for replicating the observed trajectories of the rifts to date, and suggests at possible future crack trajectories and their influence on the ice shelf. The manuscript is generally well written and organized, and is likely to find broad interest among ice shelf glaciologists. I really only have minor comments and suggestions for the manuscript to clarify certain aspects and elaborate on others.

The abstract notes that a "simple fracture propagation criterion" was successfully used to hindcast the rift trajectories and suggest future trajectories. The term "fracture criterion" is commonly used to represent something like "stress intensity exceeding fracture toughness" from fracture mechanics or "stress exceeding strength" from strength of materials theory. What has actually been applied is a heuristic for iteratively and manually lengthening fractures based on (visual?) inspection of the orientation of the first principal stress ahead of the current crack tip location. I actually think this is quite a clever method, so I am not criticizing it here. Rather, I think it is somewhat misleading to call it a "fracture criterion" or "fracture model" in the usual sense of these terms. Perhaps I am being overly picky about nomenclature here, but after reading the abstract I was expecting to see more of a "hands off" fracture propagation model based on my understanding of the usual use of the term "fracture criterion."

In general I think a bit more detail is needed on the modeling aspects of this study. In particular, more information is needed on the inverse modeling and fracture propagation algorithm. Specifically:
- Is regularization used in the inversion? If so, how much? And how is the level of regularization determined? The pattern of stresses will vary depending on level of regularization applied (which is often at least partially a subjective decision). Since you are relying on the stress orientations for predicting fracture trajectory, some details here are needed.
- It might be helpful to see an example of a single step of the crack propagation algorithm to better outline this heuristic procedure. For example, show the current extent of the crack, and the stress rosettes ahead of the crack, at a scale that resolves the step size (chosen as the mesh resolution here). Ahead of an existing crack there will be multiple rosettes, in principle one rosette for every triangular finite element (since you are using linear elements). Unless there is a rosette in an exact straight-line distance ahead of the current rift tip, I presume some choice must be made about some kind of averaging of the principal stress directions from rosettes in the vicinity of the tip. Is this simply a choice based on visual inspection? The directions may not vary much in this particular case study, but if the rift encounters a region of strongly varying flow then the decision about what direction to manually project the rift may not be so straightforward.
- P15 L30 (and subsequent residual reporting): I don't think the "mean residual" is the best measure of model performance here, since you are looking at a misfit that can have positive and negative values. You could have areas of very large negative and positive misfit (indicating a potentially poor fit to the observed data) and yet still have a mean/median residual close to

zero. In this case it would be better to report something like the root mean square or mean absolute deviation.

- Figure 8: The stress rosettes are too small to see! The areas around the rifts could be zoomed in on much more, as it is not necessary here to see the full extent of previous figures. As the figure stands currently, I cannot really judge the performance of your manual rift trajectories. It is clear that they are quite close to the actual trajectories, but I think most readers would like to better see the detail around the individual cracks than the whole ice shelf domain.

Additional specific and line-by-line comments:

- The use of the term "chasm" sounds particularly ominous, but is a rather vague and unspecific term for a rift. This term is not defined at first use in the manuscript either, which could leave some readers in question as to what specifically a chasm is (especially as "chasm" and "rift" are used interchangeably in the text). Moreover, in the literature I don't see the term "chasm" being used to describe rifts, which in the context of an ice shelf refers specifically to a through-thickness fracture (and not other features that might also be considered "chasms" such as partial-thickness crevasses or moulins). Since the manuscript is dealing specifically with through-thickness fractures, I would recommend using the more common and specific term "rift". I have no problem with using "chasm" in a naming convention (e.g. "Chasm 1").
- Throughout the text you interchange Halley "VI" and Halley "6" for naming the research station. For example, Figure 1 shows "Halley 6" and "Halley 6a" but the caption notes "Halley VI" and "Halley VIa". Probably best to choose one style of naming and stick to it throughout.
- P2 L8: calving induced -> "calving-induced"
- P3 L8: ice-shelf wide -> "ice-shelf-wide"
- P4 L6: it could be helpful to indicate on an earlier figure the outline of this previous calving event. It is eventually shown in Figure 8, but when discussed in the early part of the manuscript here it might be helpful on e.g. Figure 1 or 4 as well.
- Figure 2 caption:
  - In describing panel (b), you mention "blue-to-purple" dots for GPR data, but I only see pink dots which seem to indicate the crack tip. Can you clarify?
  - The radius of circles in panel (b) are said to represent the "local strain rate" which presumably is measured along baselines between pairs of GPS stations across the rift (this is later described on P7). It would be worth being more specific about this in the caption here, as strain rate is formally a tensor.
  - In describing panel (c), you mention the "Gatekeeper" but I didn't see this defined.
- Describing "average propagation speed" of a rift (Figure 2 caption and elsewhere) may not be appropriate if the lengthening of the rift comes in episodic bursts (which seems to often be the case). The distribution of actual propagation speeds would be strongly bi-modal, with long periods of no propagation and short periods of very fast propagation. For such a distribution, the mean is probably not a meaningful measure of central tendency. It might be more appropriate to describe this as something like a "lengthening rate," which would of course have the same units but not the same meaning as "propagation speed."
- Figure 3: It's not clear how/why the blue dots have been connected in panel (a)
- P6 L 6-7: It is rather hard to see this in Figure 3, as the horizontal scale has presumably been adjusted to view the meteoric inclusion rather than to highlight the fracture identification. As

both points are interesting in this context, perhaps it would be appropriate to show additional panels that zoom in on the fractures in the radar sections.

- P6 L9: extend -> extent

- P7 L24-25: I'm not sure I would call the rate of opening across an existing fracture a "strain rate." You can use the same units, but this isn't really a material rate of strain in the usual sense of the term "strain rate." Ahead of the crack tip this term would be appropriate.

- Figure 4: upon reading the text describing the Halloween Crack and studying Figure 4, I was missing the context of the general flow structure of the ice shelf. Although this later comes in Figure 7, it would have been helpful at this stage to determine whether the GPS baselines are representative of the crack's influence or just the divergent flow dynamics due to the geometry of the shelf and the presence of the McDonald Ice Rumples.

- P9 L30-31: this may in fact always be the case if rifts propagate in episodic bursts, which relates to my comment above about characterizing an "average" propagation speed.

- I struggled with Figure 5, as I didn't yet have the context of the flow vectors shown in Figure 7. At this point I wasn't sure whether the Halloween Crack was an opening-mode crack or a shear crack (the situation is more obvious for Chasm 1). The distinction is important for interpreting this figure. If it has any component of shear, then the change in baseline is in part due to shear translation across the crack. Clearly from panel (b) the appearance and influence of the crack is obvious. However, I think it would be helpful to resolve the GPS signals into components transverse to and parallel to the crack, after which the specific components of shear translation and crack opening can be resolved. Using a simple baseline connecting stations across this crack does not resolve this.

- Figure 5 panel (c) (and in text on P 11): This baseline is across the McDonald Ice Rumples and thus beyond the extent of the Halloween Crack, so I don't see why they would have anything to do with the crack. The change in baseline looks to me like it is due to the flow diverging around the ice rumples, which would happen with or without the crack. Thus I'm not convinced that this panel is useful here. I suspect that it is simply coincidence that the change from linear to nonlinear baseline rate-of-change (blue to red in the panel) happens around the same time the crack forms.

- P10 L2: as per comment above about resolving shear vs. opening displacement, I'm not convinced that the baseline change is due solely to "opening of the crack"

- P11 L12: as above, "spreading rate" may have both opening and shear components

- P11 L14: but you don't have GPS data for the K0-I0 baseline on the date that the crack formed, so I don't see how you can substantiate this claim. The extension rate may have sharply increased, but can this not be (at least in part) due to changing flow dynamics from these units diverging around the ice rumples?

- P11 L20: I'm convinced on the influence of the crack for H0, and probably at least partially for I0. However, does the speed normally increase in the region of I0 as the ice shelf moves past the ice rumples and begins to flow more or less unobstructed? A version of e.g. Figure 7a from before the Halloween Crack formed would be helpful for addressing this question.

- P13 L10: high strain -> high strain rate

- P14 L20 (and elsewhere): principle stress -> principal stress

- P14 L25: do you mean Sentinel-1 here?

---

## Referee Comment (RC2) · Anonymous Referee #2 · 6 Nov 2017

The authors present a collection of detailed observation data regarding the recent formation and propagation of two rifts in the BIS which may be crucial for its future. The first, called Chasm 1 by the authors, emerged and has been reactivated from a pattern of curved cracks and now propagates towards MIR. The second one is the Halloween crack which has been initiated near the MIR, and moves away from this spot rapidly. Based on numerical calculations in conjunction with the observed kinematical data, a prediction of the future rift propagation trajectories is made.

The observation data are of great value, as they provide, among others, a detailed insight into the kinematics of the crack propagation, the widening and the deformations near the rift tips. Also the numerical crack path predictions are interesting and I am sure that they will attract considerable attention. The paper is well written and clearly

in most parts. In addition to these rather general remarks, I would like to make a few comments on points which have struck me.

- The main point is the following: I am not absolutely sure, but there seems to be an inconsistency between the velocity field as shown in Fig.7a and the obvious assumption that both cracks are pure mode I cracks. Figure 7a shows clear velocity jumps across the entire rifts in their normal as well as in their tangential direction which suggests a so-called mixed-mode loading. Such a mixed-mode loading would usually lead (homogeneity of the ice shelf presumed) to a crack curving (in this case to the left in propagation direction) and not to a nearly straight crack as predicted from the model. I wonder what the reasons for this apparent contradiction are. It might be that the velocity field is not sufficiently correct. But, on the other hand, this field has been used to optimize the model, i.e. the model results should show the same tendency. Perhaps the authors can resolve this question easily.

- As the authors mention on page 15, the calculated principal stress trajectories have been used for crack path prediction instead of the measurable principal strain-rate trajectories because of the smoothening filter effect of the field equations. Here, a few explanations would be helpful, how this filter works and how its results depend on the input data as e.g. the measured velocity field. I wonder to what extent the filter may smoothen important observation facts like the velocity jumps.

- On page 18 the authors mention that their propagation algorithm makes use of the direction of maximum tensile stress, but that the stress magnitudes are not a good indicator for the formation of new rifts or for predicting the future propagation. This statement might be correct, but this is not directly shown in this paper. In this context it should be emphasized that usual fracture criteria never use directly the stress magnitude and that the question of an appropriate fracture criterion is not touched in this paper.

- Some more details regarding the used inversion method or the resolution of the

measured velocity field and calculated stress field would have been helpful and interesting for the reader. For example, if available, one could use sufficiently accurate displacement-, velocity- or stress fields to check the applicability of different fracture criteria.

- In Fig 2b caption and page 7: Though clear from the context, the term 'local strain rate network' may mislead some readers. Measured are not strains but relative displacements (widening per length). Furthermore, the white marked stakes are hardly distinguishable from the yellow O0/N0 markers. In R18: combing -> combining

- In Fig.7b and page 19: Again, though clear from the context, the term 'effective strain' may mislead some readers. In addition, it would have been more informative not to use x,y-coordinates, but local normal and tangential directions.

- Fig.3b: I see the drawn black lines, but I cannot see the cracks (magnification problem?)

- P14, R24: '. . .based on the observed stresses. . ..': stresses have not been observed and cannot be observed directly! They may be calculated.

- Multiple: 'principle stress' -> principal stress

- P15, R30-35: the details of the principal stress rosette pattern in the vicinity of MIR can only be seen if the figures are sufficiently magnified. It is indeed 'radial' but, strictly speaking, not symmetric since the radial principal stress changes its sign during circulating MIR. The pattern corresponds to that of a point force in a plate acting in the direction of the HC.

P3, R20-21: I agree fully with the opinion that observation data in conjunction with numerical simulations, as they are impressively presented by the authors, will improve the understanding of rift propagation and of the accuracy of relevant predictions. But I doubt the need for specific iceberg calving laws. The calving law is already known and given by the laws of continuum and fracture mechanics. The only and very difficult

problem is to describe the rift evolution until final separation.

---

## Author Comment (AC1) · 6 Dec 2017

**Response to RC1**

**Jan De Rydt, G. Hilmar Gudmundsson, Thomas Nagler, Jan Wuite, Edward C. King**

**4 December 2017**

This manuscript reports on detailed observations and model analysis of two rifts in the Brunt Ice Shelf. These rifts may threaten the structural integrity of the shelf, and at the very least have caused quite some concern for the fate of the Halley Research Station on the shelf. An impressive array of observations have been collected on the behaviour of these rifts, and the observations alone reveal quite a lot about the nature of rift propagation. A numerical model is applied to determine the orientation of principal stresses in the shelf, and a clever heuristic algorithm for manually modeling the trajectories of the cracks is applied. This heuristic proves successful for replicating the observed trajectories of the rifts to date, and suggests at possible future crack trajectories and their influence on the ice shelf. The manuscript is generally well written and organized, and is likely to find broad interest among ice shelf glaciologists. I really only have minor comments and suggestions for the manuscript to clarify certain aspects and elaborate on others.

We thank the reviewer for these positive comments, and for a very thorough review that has allowed us to improve the manuscript in many places. Individual replies are formulated below, and all updated figures are listed at the end of this document.

The abstract notes that a "simple fracture propagation criterion" was successfully used to hindcast the rift trajectories and suggest future trajectories. The term "fracture criterion" is commonly used to represent something like "stress intensity exceeding fracture toughness" from fracture mechanics or "stress exceeding strength" from strength of materials theory. What has actually been applied is a heuristic for iteratively and manually lengthening fractures based on (visual?) inspection of the orientation of the first principal stress ahead of the current crack tip location. I actually think this is quite a clever method, so I am not criticizing it here. Rather, I think it is somewhat misleading to call it a "fracture criterion" or "fracture model" in the usual sense of these terms. Perhaps I am being overly picky about nomenclature here, but after reading the abstract I was expecting to see more of a "hands off" fracture propagation model based on my understanding of the usual use of the term "fracture criterion."

We acknowledge the reviewer's comment and have reformulated this part of the abstract. As there are abundant potential pitfalls in fracture mechanics nomenclature, we have reverted to a more descriptive approach. Further to the reviewer's remark we should point out that, albeit heuristic, our algorithm for predicting rift trajectories is not based on a 'manual' or 'visual' approach, but is a fully automatic/programmed procedure that uses a stress field on a discretized grid, an initial fracture location, and a propagation step size as its only inputs.

The relevant part of the abstract has been reformulated as follows:

"A numerical ice-flow model and a simple propagation algorithm based on the stress distribution in the ice shelf were successfully used to hindcast the observed trajectories"

In general I think a bit more detail is needed on the modeling aspects of this study. In particular, more information is needed on the inverse modeling and fracture propagation algorithm. Specifically:

• Is regularization used in the inversion? If so, how much? And how is the level of regularization determined? The pattern of stresses will vary depending on level of regularization applied (which is often at least partially a subjective decision). Since you are relying on the stress orientations for predicting fracture trajectory, some details here are needed.

The inversion procedure uses a Tikhonov regularization for the rate factor in Glen's flow law. The optimal regularization multiplier,  $\lambda$ , was determined through an L-curve approach, as shown in Figure R1 below. A value of  $\lambda = 10^3$  was chosen to optimize the misfit and to avoid overfitting. In areas away from the grounding line and the immediate vicinity of the MIR, the solution converged quickly (within 100 iterations) towards observed values of the surface velocities, and the inversion was ended after 200 iterations.

Figure R1. L-curve approach to determine the optimal regularization multiplier, shown as labels next to each point on the curve. The value  $10^3$  was chosen as the optimal value.

Figure R2 shows the modeled trajectories of the HC for different amounts of regularization. The trajectory for  $\lambda = 10^7$  can be rejected based on the L-curve. For  $\lambda

---

## Author Comment (AC2) · 6 Dec 2017

**Response to RC2**

**Jan De Rydt, G. Hilmar Gudmundsson, Thomas Nagler, Jan Wuite, Edward C. King**

**4 December 2017**

The authors present a collection of detailed observation data regarding the recent formation and propagation of two rifts in the BIS which may be crucial for its future. The first, called Chasm 1 by the authors, emerged and has been reactivated from a pattern of curved cracks and now propagates towards MIR. The second one is the Halloween crack which has been initiated near the MIR, and moves away from this spot rapidly. Based on numerical calculations in conjunction with the observed kinematical data, a prediction of the future rift propagation trajectories is made.

The observation data are of great value, as they provide, among others, a detailed insight into the kinematics of the crack propagation, the widening and the deformations near the rift tips. Also the numerical crack path predictions are interesting and I am sure that they will attract considerable attention. The paper is well written and clearly in most parts. In addition to these rather general remarks, I would like to make a few comments on points which have struck me.

We thank the review for these positive remarks and a constructive assessment of our work. Individual replies are formulated below, and all updated figures are listed at the end of this document.

- The main point is the following: I am not absolutely sure, but there seems to be an inconsistency between the velocity field as shown in Fig.7a and the obvious assumption that both cracks are pure mode I cracks. Figure 7a shows clear velocity jumps across the entire rifts in their normal as well as in their tangential direction which suggests a so-called mixed-mode loading. Such a mixed-mode loading would usually lead (homogeneity of the ice shelf presumed) to a crack curving (in this case to the left in propagation direction) and not to a nearly straight crack as predicted from the model. I wonder what the reasons for this apparent contradiction are. It might be that the velocity field is not sufficiently correct. But, on the other hand, this field has been used to optimize the model, i.e. the model results should show the same tendency. Perhaps the authors can resolve this question easily.

As the reviewer points out, it is difficult to discern a pure mode I from a mixed mode fracture based on the subsampled velocity data in Figure 7a. To provide evidence for a pure mode I, or at least a negligibly small mode II component, we have made the following changes to the manuscript:

- In Figure 2b we are showing strain rosettes around the tip of Chasm 1. These have been calculated using a network of 12 stakes, which have been monitored monthly using high precision GPS methods. The results provide strong evidence for extension across Chasm 1 rather than shear, i.e., a pure mode I loading.
- In Figure 4 a strain rosette was added based on the differential motion of GPS stations K0, L2 and H0 after the formation of the HC. Also here the principle

strain rate directions are oriented perpendicular and parallel to the rift, indicating limited shear.

- As the authors mention on page 15, the calculated principal stress trajectories have been used for crack path prediction instead of the measurable principal strain-rate trajectories because of the smoothening filter effect of the field equations. Here, a few explanations would be helpful, how this filter works and how its results depend on the input data as e.g. the measured velocity field. I wonder to what extent the filter may smoothen important observation facts like the velocity jumps.

We assume that the reviewer is referring to the regularization in the inversion, which determines the smoothness of the rate factor in Glen's flow law and prevents overfitting of the data. We have added the following statement to the manuscript:

"A Tikhonov regularization was used and the regularization multiplier,  $\lambda = 10^3$ , was determined through a L-curve approach to optimize the misfit and avoid overfitting. In areas away from the grounding line and the immediate vicinity of the MIR, the solution converged towards observed values of the surface velocities within 100 iterations, and the inversion was ended after 200 iterations."

- On page 18 the authors mention that their propagation algorithm makes use of the direction of maximum tensile stress, but that the stress magnitudes are not a good indicator for the formation of new rifts or for predicting the future propagation. This statement might be correct, but this is not directly shown in this paper. In this context it should be emphasized that usual fracture criteria never use directly the stress magnitude and that the question of an appropriate fracture criterion is not touched in this paper.

We have added the following statement to the introduction:

"We do not intend to address the more complex issues of fracture initialization and the propagation speed of existing rifts, but rather present a heuristic algorithm for calculating rift trajectories once an initial fracture has formed."

- Some more details regarding the used inversion method or the resolution of the measured velocity field and calculated stress field would have been helpful and interesting for the reader. For example, if available, one could use sufficiently accurate displacement-, velocity- or stress fields to check the applicability of different fracture criteria.

A detailed description of the inversion procedure is outside the scope of this work, but more information is available as part of the freely available model code for Úa (ghg@bas.ac.uk). However, changes have been made to the text in order to address the sensitivity of the fracture trajectories to model details such as the mesh resolution:

"A priori, the calculated fracture trajectories depend on the amount of regularization ( $\lambda$ ) and the model resolution. Sensitivity tests were carried out for both variables and the final trajectories of the HC were found to be independent of the exact value of  $\lambda$  and mesh resolution. A further reduction of the regularization and additional mesh

refinement did not significantly change the results, and only for a much coarser mesh (2 km nodal separation) or a much larger amount of regularization ( $\lambda > 10^5$ ) did the modeled trajectories start to deviate from the observed trajectory. It should be noted that this result is case specific, and robustness of the fracture trajectories should be considered on a case-by-case basis, in particular for applications with a more complex stress distribution."

- In Fig 2b caption and page 7: Though clear from the context, the term 'local strain rate network' may mislead some readers. Measured are not strains but relative displacements (widening per length). Furthermore, the white marked stakes are hardly distinguishable from the yellow O0/N0 markers. In R18: combing -> combining

Instead of 'local strain network' we now use 'network of snow stakes'. However, differential motion of the stakes per distance and time is a strain rate per definition; hence such networks are commonly referred to as 'strain rate network' in the literature. We have changed the color of all markers to yellow and adjusted the caption accordingly.

- In Fig.7b and page 19: Again, though clear from the context, the term 'effective strain' may mislead some readers. In addition, it would have been more informative not to use x,y-coordinates, but local normal and tangential directions.

We do not understand why the term "effective strain rate" is misleading in this context. The standard definition of this scalar quantity, at least in glaciology, is given in line 8, page 13 (old version of the ms) and plotted in Figure 7b. If the reviewer is referring to the fact that fractures are present and therefore this is not a strain rate in the strictest 'continuum mechanics' sense, then we have added the following comment to the manuscript:

"Again it should be noted that these values also incorporate widening (Mode I) and shear (Mode II) of the rifts, and they are therefore not material strain rates in the strictest sense."

- Fig.3b: I see the drawn black lines, but I cannot see the cracks (magnification problem?)

We have added two additional panels (d and e) to Figure 3 with a more detailed view of the fractures.

- P14, R24: '... based on the observed stresses. ...': stresses have not been observed and cannot be observed directly! They may be calculated.

Terminology has been changed to "calculated"

- Multiple: 'principle stress' -> principal stress

done

- P15, R30-35: the details of the principal stress rosette pattern in the vicinity of MIR can only be seen if the figures are sufficiently magnified. It is indeed 'radial' but, strictly speaking, not symmetric since the radial principal stress changes its sign during circulating MIR. The pattern corresponds to that of a point force in a plate acting in the direction of the HC.

We have adjusted the spatial extent of Figures 8a and b in order to make the strain rosettes more visible. The text already mentions that "This pattern is generated by the point-interaction between the ice shelf and the bedrock at the MIR, and has resulted in the radially-outward growth of the HC, perpendicular to the tensile stresses.". We wonder if this is different from saying that the pattern corresponds to that of a point force in a plate, as suggested by the reviewer.

P3, R20-21: I agree fully with the opinion that observation data in conjunction with numerical simulations, as they are impressively presented by the authors, will improve the understanding of rift propagation and of the accuracy of relevant predictions. But I doubt the need for specific iceberg calving laws. The calving law is already known and given by the laws of continuum and fracture mechanics. The only and very difficult problem is to describe the rift evolution until final separation.

By 'calving law' we mean the formation and propagation of ice shelf rifts in the broadest sense, i.e. from rift initialization until iceberg formation. This process will be driven by the laws of fracture mechanics, which are well understood for a broad range of materials, but are little studied for ice shelves. Particular material properties and their spatial distribution (such as 'fracture toughness') are currently unknown for ice shelves, and it is unclear what determines whether/where rifts form and at what speed they propagate. Complex full-Stokes models with implemented fracture mechanics will be of help, but ultimately a more simplified set of 'calving criteria' will have to be developed for application in large-scale ice shelf models.

**UPDATED FIGURES**

Figure 2. (a) Historical outlines of Chasm 1 from 6 January 2000 (red) to 15 March 2017 (purple), as the rift got advected by the ice flow. Outlines were obtained from a sequence of Landsat-7 and Landsat-8 panchromatic images, with acquisition dates as indicated by the black ticks in the colour bar. The background image is a subset of a Landsat-8 scene from 15 March 2017. The red dot indicates the location of Halley 6 research station, yellow dots are the locations of 4 permanent dual frequency GPS stations, and the black box outlines the extent of Figure b. The inset in the top right shows an aerial image of Chasm 1 taken in December 2015, looking from the reference point A towards the crack tip. (b) Detailed overview of the area around the tip of Chasm 1, showing the local snow stake network (vellow dots), an example of the GPR survey lines (white lines), and progression of the crack tip as obtained from satellite (blue-to-purple outlines) and GPR data (black dots). Strain rosettes were obtained from the relative movement of the snow stakes, and capture the widening of Chasm 1 as well as local strain rates in the ice shelf. (c) Propagation of the tip of Chasm 1 with respect to its historical location prior to the reactivation in December 2012, based on Landsat-7/8 images (grey markers), and monthly GPR surveys (magenta markers). A linear fit through all datapoints shows an average lengthening rate of the rift of 1.36 km/vr. (d) Baseline distance across Chasm 1 as a function of time, measured by two pairs of permanent GPS stations (00, 100, and 11, 120, 100, and five pairs of snow stakes ( $A_{i}$ ,  $B_{i}, with $i \in \{100, 200, 400, 500, 600\}$  in (b)). The least-squares quadratic fit is plotted as a dashed line.

---

## Author Response (AR1)

[revised manuscript text omitted]

---

## Author Response (AR2)

Cambridge CB3 0ET
United Kingdom

Telephone +44 (0) 1223 221656
www.antarctica.ac.uk

Dear Prof Vieli,

Many thanks for reviewing the changes and for accepting our manuscript. Please extend our thanks to both reviewers for their comprehensive assessment, valuable suggestions and insightful comments.

In reply to your minor points:

p. 6 line 10: something wrong here with plural/singular, I guess it should be in plural: '…by vertical black stripes.' (delete 'a')

> Sentence has been changed.

p. 14, line 10: something wron at sentence beginning with 'Average values…', delete 'Severa'???

> Sentence has been changed.

p. 20 line 21: to be consistent with changes in abstract and methods in response to the editor comment, it should here say 'algorithm' rather than 'criterion'.

> Criterion has been changed to algorithm.

Just checking a general point: as a response to review the term 'chasm' has been replaced by 'rift' throughout except when you refer to 'Chasm 1'. I assume you deliberately did not change this to 'Rift 1' as this is the common name of the feature.

> This is correct, we have used 'Chasm 1' as a place name, consitent with revious publications. The reviewer did not object against using Chasm as part of a name.

Kind regards,
Jan De Rydt and co-authors